# DPBLOOMFILTER: SECURING BLOOM FILTERS WITH DIFFERENTIAL PRIVACY

## ABSTRACT

The Bloom filter is a simple yet space-efficient probabilistic data structure that supports membership queries for dramatically large datasets. It is widely utilized and implemented across various industrial scenarios, often handling massive datasets that include sensitive user information necessitating privacy preservation. To address the challenge of maintaining privacy within the Bloom filter, we have developed the DPBloomfilter. This innovation integrates the classical differential privacy mechanism, specifically the Random Response technique, into the Bloom filter, offering robust privacy guarantees under the same running complexity as the standard Bloom filter. Through rigorous simulation experiments, we have demonstrated that our DPBloomfilter algorithm maintains high utility while ensuring privacy protections. To the best of our knowledge, this is the first work to provide differential privacy guarantees for the Bloom filter for membership query problems.

## 1 INTRODUCTION

In the current data-rich era, extracting meaningful information from the ever-growing volume of data presents a significant challenge (Sagiroglu & Sinanc, 2013). To address this challenge, various data structures have been developed to facilitate the extraction of insights from vast datasets (Chang, 2006), such as the Bloom filter (Bloom, 1970), count-min sketch (Cormode, 2009), hyperloglog (Flajolet et al., 2007), and so on. Among them, the Bloom filter mainly handles membership queries in big data (Bloom, 1970); count-min sketch handles the frequency of occurrence of a certain type of data in big data (Cormode, 2009); Hyperloglog is used to count the cardinality of a set of data, that is, the number of different elements in this set of data (Flajolet et al., 2007).

In this paper, we focus more on the Bloom filter (Bloom, 1970), which is a space-efficient probability data structure that deals with membership queries. Due to its efficient space utilization and low time complexity, it is widely used in various scenarios, especially industry scenarios requiring massive data processing and low-latency response capability. Classical scenarios include database systems and web-cache systems (Gremillion, 1982; Najork et al., 2009; Mun & Lim, 2016; Patgiri et al., 2020).

In addition to the scenarios mentioned above, the Bloom filter is also used in various scenarios involving sensitive user data. One usage is the privacy-preserving dataset intersection: When two organizations want to find out what user data they have in common without revealing specific user information, Bloom filters can be used. By converting the respective user datasets into Bloom filters and then performing an intersection operation, common elements can be determined without exposing specific user records (Budhkar, 2013; Jeffrey & Steffan, 2011). Another scenario is anonymous login: Bloom filters can store hash values of login credentials. When a user tries to log in, the system can check whether the hash of the credentials may exist in the filter instead of storing the actual password hash (Laufer et al., 2011; Berardi et al., 2020). Since the content inserted into the Bloom filter is user-sensitive, preventing attackers from reconstructing user-sensitive information from the released Bloom filter vector is an essential task.

In this work, we consider the differential privacy of the Bloom filter under the membership query scenario. The membership query problem involves storing information about a set of elements $S$ in a space-efficient manner to determine if an element $x$ is a member of $S$. One example is the membership query application of the Bloom filter in streaming media recommendation (Wang et al.,

2014), such as Tiktok. That is, the Bloom filter will be used for filtering to prevent users from being recommended duplicate content when using streaming media. The Bloomfilter vector mentioned above will also be released to other businesses, such as advertising, e-commerce, etc. When the Bloomfilter vector is released, the user's privacy information, which videos the user has watched, needs to be well protected.

Thus, we introduce our DPBloomfilter (Algorithm 1) to protect the sensitive user information stored in the Bloomfilter vector, i.e. the $m$ index binary bits based on the hash values generated by $k$ different hash functions. To implement a differential privacy budget, we used the classic random response technique (Warner, 1965) (Definition 3.4 ) in differential privacy, which randomly flips some bits to ensure that attackers cannot restore sensitive user data from neighboring datasets (Definition3.2). We theoretically show that our DPBloomfilter achieves $(\epsilon, \delta)$-DP guarantee, where the main technique is that we first ensure each bit holds a certain DP guarantee so that we achieve $(\epsilon, \delta)$-DP for the entire Bloom filter. Also, we have theoretically proved that our DPBloomfilter has high utility when DP parameters are in a certain regime. Furthermore, our empirical evidence verifies our utility analysis that our DPBloomfilter can procedure membership query services with high accuracy while protecting user data privacy. While providing privacy guarantees, our algorithm preserves the same running complexity as the standard Bloom filter.

Our contribution can be summarized as follows: (1) To the best of our knowledge, this is the first work to provide DP for the Bloom filter for membership query problems. (2) We have proved from a theoretical perspective that DPBloomfilter can achieve $(\epsilon, \delta)$-DP under the random response mechanism while preserving the same running time complexity compared with the standard Bloom filter. (3) We have proved from a theoretical perspective that when the DP parameters $\epsilon$ and $\delta$ are very small, DPBloomfilter can still maintain good utility. (4) Our simulation experiments also reflect the same effect as our theoretical results. The two confirm each other.

## 2 RELATED WORK

### 2.1 BLOOM FILTER

The Bloom filter is first introduced by (Bloom, 1970) and there are many variants of the Bloom filter. One variant is the Cuckoo filter (Fan et al., 2014), which "kicks out" the old hash value to another place when a hash conflict occurs. This implementation principle enables it to support the probability data structure of membership queries with deletion operation. Compared with the Standard Bloom filter, it is more suitable for application scenarios with frequent element updates, such as network traffic monitoring (Grashöfer et al., 2018) and cache system (Wang et al., 2022).

Another variant is the Quotient filter (Geil et al., 2018), which differs from the traditional Bloom filter. It implements the heretical storage form of hash value atmosphere quotient and remainder. This approach results in the Quotient filter requiring less storage space and offering faster query speeds than the standard Bloom filter. It is more suitable for membership queries in scenarios with limited resources and high latency requirements (Pandey et al., 2021; Al-Hisnawi & Ahmadi, 2016).

### 2.2 DIFFERENTIAL PRIVACY

Differential privacy is a technique used to defend against privacy attacks, first proposed by Dwork et al. (Dwork et al., 2006). It has become one of the most popular frameworks for ensuring privacy in theoretical analysis and a wide range of application scenarios (Li et al., 2017; Yang et al., 2023; Wang et al., 2023; Cheng et al., 2024; Sajadmanesh & Gatica-Perez, 2024; Gu et al., 2025; Li et al., 2024a;c;b; Liang et al., 2024; Fan et al., 2024; Song et al., 2023; Liu et al., 2024; Hu et al., 2024; Yu et al., 2024). Gaussian mechanism (Dwork et al., 2006) and Laplace mechanism (Dwork et al., 2014) of DP are widely used techniques to achieve privacy budget. These two mechanisms control the amount of privacy provided by adjusting the variance of the added noise. However, these two mechanisms are very useful when the output is continuous, but they are slightly weak when the output is discrete. However, another classic way to make a data structure private is to add a random responses mechanism (Warner, 1965), also called a "flip coin". Specifically, some discrete values in the data structure are flipped with a certain probability to achieve privacy (Li & Li, 2023; 2024). By controlling the probability of flipping, a given privacy budget is achieved. Over the past decade, numerous works have applied differential privacy to data structures or deep

learning models. (Kasiviswanathan et al., 2013) applied differential privacy to graph data structure and designed differentially node-private algorithms by projecting input graphs onto bounded-degree graphs, enhancing privacy while maintaining accuracy in realistic network analyses. (Wang et al., 2018) introduced an adaptive method for directly collecting frequent terms under local differential privacy by constructing a tree, which can overcome challenges of accuracy and utility compared to existing n-gram approaches. (Fletcher & Islam, 2019) focused on applying differential privacy to classical data mining data structures, specifically decision trees, and analyzed the balance between privacy and the utility of existing methods. (Zhao et al., 2022) demonstrated the integration of differential privacy into linear sketches, ensuring privacy while maintaining high performance in processing sensitive data streams. A related work (Alaggan et al., 2012) introduced the BLIP mechanism, which also applies the Random Flip mechanism to the Bloom Filter. Here, we outline the differences between our work and (Alaggan et al., 2012) as follows: (1) Our proposed DPBloom-Filter can satisfy $(\epsilon, \delta) - DP$, while (Alaggan et al., 2012) only verified that BLIP mechanism can satisfy $\epsilon$-DP; (2) (Alaggan et al., 2012) did not provide theoretical guarantees for the utility of the BLIP mechanism.

**Roadmap.** Our paper is organized as follows: Section 3 presents the preliminary of Bloom Filter and Differential Privacy. In Section 4, we outline the main results of our algorithm. In Section 6, we elaborate on the underlying intuitions that informed the design of the DPBloomfilter. In Section 7, we conclude our paper.

## 3 PRELIMINARY

**Notations.** For any positive integer $n$, let $[n]$ denote the set $\{1, 2, \cdots, n\}$. We use $\mathbb{E}[]$ to denote the expectation operator and $\Pr[]$ to denote probability. We use $n!$ to denote the factorial of integer $n$. We use $A_m^n := \frac{m!}{(m-n)!}$ to denote the number of permutation ways to choose $n$ elements from $m$ elements considering the order of selection. We use $\binom{m}{n} := \frac{m!}{n!(m-n!)}$ to denote the number of combination ways to choose $n$ elements from $m$ elements without considering the order of selection. We use $F_X(x)$ to denote the Cumulative Distribution Function (CDF) of a random variable $X$ and use $F_X^{-1}(1 - \delta)$ to denote the $1 - \delta$ quantile of $F_X(x)$.

### 3.1 BLOOM FILTER

A Bloom filter is a space-efficient probabilistic data structure used to test whether an element is a member of the set. Its formal definition is as follows.

**Definition 3.1** (Bloom Filter, (Bloom, 1970)). *A Bloom filter is used to represent a set $A = \{x_1, x_2, \ldots, x_{|A|}\}$ of $|A|$ elements from a universe $U = [n]$. A Bloom filter consists of a binary array $g \in \{0, 1\}^m$ of $m$ bits, which are initially all set to 0, and uses $k$ independent random hash functions $h_1, \ldots, h_k$ with range $\{0, \ldots, m - 1\}$. These hash functions map each element in the universe to a random number uniform over the range $\{0, \ldots, m - 1\}$ for mathematical convenience. The computation time per execution for all hash functions is $\mathcal{T}_h$. Bloom Filter supports the following operations: (1) INIT(A). It takes dataset $A$ as input. For each element $x \in A$, the bits $h_i(x)$ of array $g$ are set to 1 for $1 \leq i \leq k$. (2) QUERY($y \in [n]$). It takes an element $y$ as input. If all $h_i(y)$ are set to 1, then it outputs a binary answer to indicate that $y \in A$. If not, then it outputs $y$ is not a member of $A$.*

A Bloom Filter does not have false negative issues but may yield a *false positive* issue, where it suggests that when a query is made to check if an element is in the set but all the positions it maps to are already set to 1 (due to previous insertions of elements of dataset $A$). Following previous literature (Li & Li, 2023; Broder et al., 1998; Li & König, 2011; Li et al., 2012), we assume a hash function selects each array position with equal probability. Then, the false positive rate of the Bloom Filter defined above can be mathematically approximated by the formula as $(1 - e^{-\frac{k|A|}{m}})^k$.

## 3.2 Differential Privacy

We begin with introducing the neighboring dataset. We follow the standard definition in the DP literature of "neighboring" for binary data vectors: two datasets are adjacent if they differ in one element. The formal statement is as follows.

**Definition 3.2** (Neighboring Dataset, (Dwork et al., 2006)). *$A, A' \in \{0, 1\}^n$ are neighboring datasets if they only differ in one element, i.e., $A_i \neq A_i'$ for one $i \in [n]$ and $A_j = A_j'$, for $j \neq i$.*

Differential Privacy (DP) ensures that the output of an algorithm remains statistically similar under neighboring datasets introduced above, thereby protecting individual privacy. Its formal definition is as follows.

**Definition 3.3** (Differential Privacy, (Dwork et al., 2006)). *For a randomized algorithm $M : U \to Range(M)$ and $\epsilon, \delta \geq 0$, if for any two neighboring data $u$ and $u'$, it holds for $\forall Z \subset Range(M)$, $\Pr[M(u) \in Z] \leq e^\epsilon \Pr[M(u') \in Z] + \delta$, then algorithm $M$ is said to satisfy $(\epsilon, \delta)$-differentially privacy. If $\delta = 0$, $M$ is called $\epsilon$-differentially private.*

Finally, we introduce the formal definition of the random response mechanism.

**Definition 3.4** (Random response mechanism). *Let $g \in \{0, 1\}^m$ denote the $m$ bit array in the Bloom filter. For any $j \in [m]$, let $\widetilde{g}[j]$ denote the perturbed version of $g[j]$, using the random response mechanism. Namely, for any $j \in [m]$, we have*

$$\Pr[\widetilde{g}[j] = y] = \begin{cases} e^{\epsilon_0}/(e^{\epsilon_0} + 1), & y = g[j] \\ 1/(e^{\epsilon_0} + 1), & y = 1 - g[j] \end{cases}$$

Let $a = e^{\epsilon_0}/(e^{\epsilon_0} + 1), b = 1/(e^{\epsilon_0} + 1)$. Since $a/b = e^{\epsilon_0}$, this implies random response can achieve $\epsilon_0$-DP.

# 4 Main Results

In Section 4.1, we will provide the privacy of our algorithm. Then, we will examine the utility implications of our algorithm applying a random response mechanism. In Section 4.2, we introduce the utility guarantees of our algorithm. In Section 4.3, we demonstrate that DPBloomfilter does not import the running complexity burden to the standard Bloom filter.

## 4.1 Privacy for DPBloomfilter

Algorithm 1 illustrates the application of the random response mechanism to the standard Bloom filter, thereby accomplishing differential privacy. In detail, once the Bloom filter is initialized, each bit in the $m$-bit array is independently toggled with a probability of $\frac{1}{\epsilon_0 + 1}$. Our algorithm will ensure that modifications to any element within the dataset are protected to a degree, as the DPBloomfilter maintains the privacy of the altered element. Then, we present the Theorem demonstrating that our algorithm is $(\epsilon, \delta)$-DP.

**Theorem 4.1** (Privacy for Query, informal version of Theorem C.2). *Let $N := F_W^{-1}(1 - \delta)$ and $\epsilon_0 = \epsilon/N$. Then, we can show, the output of* QUERY *procedure of Algorithm 1 achieves $(\epsilon, \delta)$-DP.*

Theorem 4.1 shows that our DPBloomfilter in Algorithm 1 is $(\epsilon, \delta)$-DP. Our main technique leverages the single-bit random response technique to enhance the privacy properties of the traditional Bloom filter by composition rule (Lemma A.1).

## 4.2 Utility for DPBloomfilter

Despite the introduction of privacy-preserving mechanisms, our algorithm still ensures that the utility of the Bloom Filter remains acceptable. This is achieved through careful calibration of the Random Response technique parameters, balancing the need for privacy with the requirement for accurate set membership queries. Here, we present the theorem for the entire utility loss between the output of our algorithm and ground truth.

---

**Algorithm 1** Differentially Private Bloom Filter

---

1: **data structure** DPBLOOMFILTER                                      ▷ Theorem 4.1, 4.2, 4.3
2:
3: **members**
4:     $[n]$ is the set universe
5:     $k$ is the number of hash functions
6:     Let $g \in \{0,1\}^m$.
7:     Let $h_i : [n] \to [m]$ for each $i \in [k]$
8: **end members**
9:
10: **procedure** INIT($A \subset [n], k \in \mathbb{N}_+, m \in \mathbb{N}_+$)                ▷ Lemma E.1
11:     Let $m$ denote the length of the filter
12:     We pick $k$ random hash functions, say they are $h_1, h_2, \cdots, h_k$, for each $i \in [k]$, $h_i : [n] \to$
        $[m]$
13:     Set every entry of $g$ to 0.
14:     Let $N = F^{-1}(1 - \delta)$, and $\epsilon_0 := \epsilon/N$
15:     **for** $x \in A$ **do**
16:         **for** $i = 1 \to k$ **do**
17:             Let $j \leftarrow h_i[x]$
18:             $g[j] \leftarrow 1$
19:         **end for**
20:     **end for**
21:     **for** $j = 1 \to m$ **do**
22:         $\widetilde{g}[j] \leftarrow g[j]$ with probability $\frac{e^{\epsilon_0}}{e^{\epsilon_0}+1}$
23:         $\widetilde{g}[j] \leftarrow 1 - g[j]$ with probability $\frac{1}{e^{\epsilon_0}+1}$
24:     **end for**
25: **end procedure**
26:
27: **procedure** QUERY($y \in [n]$)                    ▷ Lemma E.2, Theorem 4.1, Theorem 4.2
28:     **for** $i = 1 \to k$ **do**
29:         Let $j \leftarrow h_i[y]$
30:         **if** $\widetilde{g}[j] \neq 1$ **then**
31:             **return** false
32:         **end if**
33:     **end for**
34:     **return** true
35: **end procedure**
36:
37: **data structure**

---

**Theorem 4.2** (Accuracy (compare DPBloom with true-answer) for Query, informal version of Theorem D.4). *Let $z \in \{0,1\}$ denote the true answer for whether $x \in A$. Let $\widehat{z} \in \{0,1\}$ denote the answer for whether $x \in A$ output by Bloom Filter. Let $\alpha := \Pr[z = 0] \in [0,1]$, $t := e^{\epsilon_0}/(e^{\epsilon_0}+1)$, and $\delta_{\mathrm{err}} > 0$. Then, we can show*

$$\Pr[\widetilde{z} = z] \geq \delta_{\mathrm{err}} \cdot \alpha \cdot (1 - t - t^k) + \alpha \cdot t.$$

Theorem 4.2 shows that when most queries are not in $A$, the above theorem can ensure that the utility of DPBloomfilter has a good guarantee. Namely, in such cases, answers from DPBloomfilter are correct with high probability.

### 4.3 RUNNING COMPLEXITY OF DPBLOOMFILTER

Now, we introduce the running complexity for the DPBloomfilter in the following theorem.

**Theorem 4.3** (Running complexity of DPBloomfilter). *Let $\mathcal{T}_h$ denote the time of evaluation of function $h$ at any point. Then, for the DPBloomfilter (Algorithm 1) we have*

- *The running complexity for the initialization procedure is $O(|A| \cdot k \cdot \mathcal{T}_h + m)$.*

- *The running complexity $O(k \cdot \mathcal{T}_h)$ for a single query.*

*Proof.* It can be proved by combining Lemma E.1 and E.2. $\qquad\square$

Our Theorem 4.3 shows that DPBloomfilter not only addresses the critical need to protect the privacy of elements stored with Bloom filter but also ensures that the data structure's utility remains acceptable, with minimal impact on its computational efficiency. By keeping the running time within the same order of magnitude as the standard Bloom filter, our approach is practical for real-world applications requiring fast and scalable set operations.

## 5 TECHNICAL OVERVIEW

In this section, we provide an overview of the techniques we used in proving our theoretical results.

### 5.1 PRIVACY GUARANTEES OF SINGLE BIT

To accomplish differential privacy, Algorithm 1 applies a random response mechanism to each bit of the standard Bloom Filter. In this section, we aim to examine the privacy guarantees for a single bit of our algorithm.

Recall that in Definition 3.1, for dataset $A \subset [n]$, we use $g[j]$ to denote the $j$-th element of array output by standard Bloom Filter. Here, we use $\widehat{g}[j]$ to denote the $j$-th element of array output by DPBloomfilter. Similarly, for any neighboring dataset $A' \subset [n]$, we use $g'[j]$ and $\widehat{g}'[j]$ to denote the $j$-th element of array output by standard Bloom Filter and DPBloomfilter. To examine the privacy guarantees for the $i$-th bit, we must consider two distinct cases.

**Case 1**. Suppose $g'[j] = g[j]$, then we can obtain (See also Lemma C.1) that for all $v \in \{0, 1\}$, we have $\frac{\Pr[\widetilde{g}[j]=v]}{\Pr[\widetilde{g}'[j]=v]} = 1$.

**Case 2**. Suppose $g'[j] \neq g[j]$, then we can obtain (See also Lemma C.1) that for all $v \in 0, 1$, we have $e^{-\epsilon_0} \leq \frac{\Pr[\widetilde{g}[j]=v]}{\Pr[\widetilde{g}'[j]=v]} \leq e^{\epsilon_0}$.

By combining the above two cases, we can demonstrate the privacy guarantees of a single bit for our algorithm.

**Lemma 5.1** (Differential Privacy for single Bit, informal version of Lemma C.1). *Let $\epsilon_0 \geq 0$ and $\widetilde{g}[i] \in \{0, 1\}$ be the $i$-th element of array output by DPBloomfilter. Then, we can show that, for all $j \in [m]$, $\widetilde{g}[j]$ is $\epsilon_0$-DP.*

### 5.2 PRIVACY GUARANTEES OF DPBLOOMFILTER

Here, we comprehensively analyze the DP guarantees for our DPBloomFilter. Recall that in Definition 3.1, for dataset $A$, we use $g$ to denote the array output by standard Bloom Filter. Here, we use $\widetilde{g}$ to denote the array output by DPBloomfilter. Similarly, for any neighboring dataset $A'$, we use $g'$ and $\widehat{g}'$ to denote the array output by standard Bloom Filter and DPBloomfilter, respectively. Here, we consider the set of indices $j$ within the range $m$ where the value of $g[j]$ and $g'[j]$ differs, which is defined as $S := \{j \in [m] : g[j] \neq g'[j]\}$. Thus, the set of indices $j$ where the value of $g[j]$ and $g'[j]$ are the same can be defined as $\overline{S} := [m] \backslash S$.

We can use the result of privacy guarantees of a single bit in Section 5.1, for any $j \in S$ and $v \in \{0, 1\}$, we have $\frac{\Pr[\widetilde{g}[j]=v]}{\Pr[\widetilde{g}'[j]=v]} = 1$, and for any $j \in \overline{S}$ and $v \in \{0, 1\}$, we have

$$e^{-\epsilon_0} \leq \frac{\Pr[\widetilde{g}[j] = v]}{\Pr[\widetilde{g}'[j] = v]} \leq e^{\epsilon_0}.$$

By applying the composition lemma (refer to Lemma A.1) , we obtain the following for any $Z \in \{0, 1\}^m$,

$$|\ln \frac{\Pr[\widetilde{g} = Z]}{\Pr[\widetilde{g}' = Z]}| \leq |S|\epsilon_0. \tag{1}$$

Here, we define $W := |S|$ for convenience. To get a better bound for Equation 1, we need to calculate the probability distribution function of the random variable $W$. Before that, we need to define two random variables we will use. Firstly, we define $Y$ as the set of distinct values among the $k$ hash values generated by the standard Bloom filter considering one $x \in [n]$. Then we consider two data $x, x' \in [n]$. We define $Z$ as the set of distinct values in $Y_x \cup Y_{x'}$.

Then firstly we proceed to calculate the distribution of $|Y|$ (see details in Lemma B.4), we can show for any $y = 1, 2, \ldots, k$

$$\Pr[|Y| = y] = \begin{cases} 1/m^{k-1}, & y = 1 \\ \binom{m}{y}(\frac{y}{m})^k - \binom{m-i}{y-i}\sum_{i=1}^{k-1}\Pr[Y = i], & y = 2, \cdots, k \end{cases}$$

Given the probability of $|Y|$, we can calculate the conditional probability of $|Z|$ conditioned on $|Y_x| = a$ and $|Y_{x'}| = b$, where $a, b \in [k]$ (see details in Lemma B.5)

$$\Pr[|Z| = z||Y_x| = a, |Y_{x'}| = b]$$
$$= \frac{A_m^a \cdot \binom{b}{t} \cdot A_{m-a}^t \cdot A_a^{b-t}}{A_m^a \cdot A_m^b}.$$

Finally, we use the property of union probability. We can calculate the probability of $W$ (see details in Lemma B.6). Recall the notations in Section 3, we use $F_X^{-1}$ to denote the $1 - \delta$ quantile of the Cumulative Distribution Function $F_X(x)$ of random variable $X$. Here, we define $N := F_W^{-1}(1 - \delta)$ Hence, by the properties of the quantile function, we have $\Pr[N \leq W] = 1 - \delta$. By choosing the appropriate value of $\epsilon_0 = \epsilon/N$, we have $|\ln \frac{\Pr[\widetilde{g} = Z]}{\Pr[\widetilde{g}' = Z]}| \leq W\frac{\epsilon}{N}$. Then we have, with probability $1 - \delta$, $|\ln \frac{\Pr[\widetilde{g} = Z]}{\Pr[\widetilde{g}' = Z]}| \leq \epsilon$. Then, we can demonstrate the privacy guarantees for DPBloomfilter (see also Theorem 4.1).

## 5.3 UTILITY GUARANTEES OF DPBLOOMFILTER

This section will present a comprehensive analysis of the utility guarantees for DPBloomfilter. We start by introducing the following conditions for the Utility guarantee of DPBloomFilter.

**Condition 5.2.** *We need the following conditions for Utility guarantees of DPBloomfilter:*

- **Condition 1.** *Assume that a hash function selects each array position with equal probability.*

- **Condition 2.** *Let $z \in \{0, 1\}$ denote the ground truth for whether an element $y \in A$.*

- **Condition 3.** *Let $\widehat{z} \in \{0, 1\}$ denote the answer output by standard Bloom Filter for whether an element $y \in A$.*

- **Condition 4.** *Let $\widetilde{z} \in \{0, 1\}$ denote the answer output by DPBloomfilter for whether an element $y \in A$*

- **Condition 5.** *Let $\alpha := \Pr[z = 0] \in [0, 1]$*

- **Condition 6.** *Let $t := e^{\epsilon_0}/(e^{\epsilon_0} + 1)$.*

Firstly, we proceed to derive the utility of the standard Bloom Filter by calculating

$$\Pr[\widehat{z} = z] = 1 - \Pr[\widehat{z} = 1|z = 0]\Pr[z = 0].$$

The above equation comes from the fact that Bloom Filter will not introduce a false negative. After the initialization process of Bloom Filter, the probability of one certain bit is not set to $1$ is (see also Lemma D.2) $(1 - \frac{1}{m})^{|A|k} \geq e^{-2|A|k/m}$. A false positive occurs when, for all $i \in [k]$, the elements $g[h_i(y)]$ are all set to $1$ after initialization. In this case, we have:

$$\Pr[\widehat{z} = 1|z = 0] = (1 - (1 - \frac{1}{m})^{|A|k})^k \leq (1 - e^{-2|A|k/m})^k.$$

Therefore, we have $\Pr[\widehat{z} = z] \geq 1 - (1 - e^{-2|A|k/m})^k\alpha$. Further if $m = \Omega(|A|k)$ and $k = \Theta(log(\alpha/\delta_{err}))$, we have $\Pr[\widehat{z} = z] = 1 - \delta_{err} \cdot \alpha$.

**Lemma 5.3** (Accuracy for query of Standard Bloom filter, informal version of Lemma D.2). *If Condition 5.2 holds, we have*

$$\Pr[\widehat{z} = z] \geq 1 - (1 - e^{-2|A|k/m})^k \cdot \alpha.$$

*Further if $m = \Omega(|A|k)$ and $k = \Theta(\log(1/\delta_{err}))$, we have*

$$\Pr[\widehat{z} = z] \geq 1 - \delta_{\mathrm{err}} \cdot \alpha.$$

We then quantify the error introduced by applying the random response mechanism in the DPBloom-filter by calculating $\Pr[\widetilde{z} = \widehat{z}]$. Using basic probability rules, we have

$$\Pr[\widetilde{z} = \widehat{z}] = \Pr[\widetilde{z} = 1|\widehat{z} = 1]\Pr[\widehat{z} = 1]$$
$$+ \Pr[\widetilde{z} = 0|\widehat{z} = 0]\Pr[\widehat{z} = 0].$$

We can compute the following term by using the definition of DPBloomfilter in Algorithm 1 (see details in Lemma D.3)

$$\Pr[\widetilde{z} = 1|\widehat{z} = 1] = (\frac{e^{\epsilon_0}}{e^{\epsilon_0}+1})^k, \text{and } \Pr[\widetilde{z} = 0|\widehat{z} = 0] \geq \frac{e^{\epsilon_0}}{e^{\epsilon_0}+1}.$$

Here we let $\Pr[\widehat{z} = 0] = \widehat{\alpha}$, note that $\widehat{\alpha} = \alpha(1 - \delta_{\mathrm{err}})$. Hence, $\Pr[\widehat{z} = 1] = 1 - \Pr[\widehat{z} = 0] = 1 - \alpha + \alpha \cdot \delta_{err}$. Then we will have (see details in Lemma D.3)

$$\Pr[\widehat{z} = z] \geq t \cdot \alpha \cdot (1 - \delta_{err}).$$

**Lemma 5.4** (Accuracy (compare DPBloomFilter with Bloom) for Query, informal version of Lemma D.3). *If Condition 5.2 holds, we can show Then, we can show*

$$\Pr[\widetilde{z} = \widehat{z}] \geq t \cdot \alpha \cdot (1 - \delta_{err}).$$

Now, we can proceed to examine the utility guarantees of DPBloomfilter by calculating $\Pr[\widetilde{z} = z]$, i.e., comparing the output of DPBloomfilter with the ground truth for the query. By combining the result of the analysis above, we will have (see more details in Theorem D.4)

$$\Pr[\widetilde{z} = z] \geq \alpha \cdot (1 - t - t^k) \cdot \delta_{\mathrm{err}} + \alpha \cdot t.$$

Then, we demonstrated the utility guarantees of our algorithm while simultaneously ensuring privacy (see Theorem 4.2). Similar to other differential privacy algorithms, our algorithm encounters a trade-off between privacy and utility, where increased privacy typically results in a reduction in utility, and conversely. An in-depth examination of this trade-off is provided as follows.

**Remark 5.5** (Trade-off between Privacy and Utility of DPBloomfilter). *An inherent trade-off exists between the privacy and utility guarantees of our algorithm. To ensure privacy, we must lower the value of $\epsilon_0$ in Theorem 4.1. On the other hand, for utility considerations (in Theorem 4.2), we define the lower bound of $\Pr[\widetilde{z} = z]$ as $u = \alpha(1 - t - t^k)\delta_{\mathrm{err}} + \alpha t$, a reduction in $\epsilon_0$ will lead to a reduction in $t$ then finally result in a reduction in $u$. This, in turn, leads to diminished utility.*

### 5.4 RUNNING TIME OF DPBLOOMFILTER

In this section, we will analyze the running time of our DPBloomfilter. Recall in Definition 3.1, we let $\mathcal{T}_h$ denote the computation time per execution for all hash functions. To analyze the algorithm's running time, firstly, we consider the running time of initialization in Algorithm 1.

It contains two steps as follows

- **Step 1.** Let's consider the initialization of the standard Bloom Filter. For a single element $x \in A$, it needs $O(k \cdot \mathcal{T}_h)$ time to compute over $k$ hash functions. And $|A|$ elements need to be inserted. Combining these two facts, it needs $|A| \cdot k \cdot \mathcal{T}_h$ time to initialize the standard Bloom Filter.

- **Step 2.** Let's consider the "Flip each bit" part in DPBloomfilter. Since there are $m$ bits in the Bloom Filter, it takes $O(m)$ time to flip each bit. Hence, it takes $O(|A| \cdot k \cdot \mathcal{T}_h + m)$ time to run the initialization function in Algorithm 1. (see also in Lemma E.1)

Then, we consider the running time of a single query in Algorithm 1. For each query $y$, the algorithm needs $O(k \cdot \mathcal{T}_h)$ time to compute the hash values of $y$ over $k$ hash functions. Hence, it takes $O(k \cdot \mathcal{T}_h)$ time to run each query $y$ in. (see also in Lemma E.2)

By combining the two running times together, we can obtain the running time of our entire algorithm is $O(|A| \cdot k \cdot \mathcal{T}_h + m)$. This highlights the advantage of our algorithm: it matches the time complexity of a standard Bloom Filter while providing a strong privacy guarantee.

## 6 DISCUSSION

**Why Random Response but not Gaussian or Laplace Noise?**    As mentioned in Section 2, Gaussian and Laplace noise are two classical mechanisms to achieve differential privacy. The advantage of the Laplace mechanism is that its distribution is concentrated on its mean. Under the same privacy budget, it will not introduce too much noise like the Gaussian mechanism due to the long-tail nature of its distribution. The advantage of the Gaussian mechanism is that it has good mathematical properties and makes it easy to analyze the utility of private data structures. However, the above two mechanisms are not as effective as the random response (flip coin) mechanism when dealing with discrete values. Here, we consider the case where the discrete values are integers. Under certain privacy budgets, the noise added by Gaussian and Laplace mechanisms does not reach the threshold of $0.5$, resulting in attackers being able to remove the noise through rounding operations easily, and the privacy of the data structure no longer exists. In our case, each bit of the Bloom filter can only be $1$ or $0$, which is consistent with the above situation. Hence, our work only considers the random response mechanism instead of classical Gaussian and Laplace mechanisms.

**Why Flip Both $0$ and $1$?**    In our work, we apply a random response mechanism to each bit in the Bloom filter, either it is $0$ or $1$. Although this will lead to a certain probability of false negatives in the Bloom filter, we argue that it is necessary to make the Bloom filter differentially private. Let's consider what will happen if we don't apply a random response mechanism like this. Suppose we only apply random responses to bits that are $1$ in the Bloom filter and leave the bits with $0$ untouched. Following the notations used in Lemma A, we use $g \in \{0, 1\}^m$ to represent the bit array generated by inserting the original dataset into the Bloom filter and $g' \in \{0, 1\}^m$ to represent the bit array generated by inserting the neighboring dataset into the Bloom filter. We use $\widetilde{g}$ and $\widetilde{g}'$ to denote their private version, respectively. Without loss of generality, for some $j \in [m]$, we assume $g[j] = 1$ and $g'[j] = 0$. Since we only apply a random response mechanism on bits with value $1$, then $\Pr[\widetilde{g}'[j] = 1] = 0$. Therefore, we cannot calculate $\Pr[\widetilde{g}[j] = 1] / \Pr[\widetilde{g}'[j] = 1]$, since the denominator is $0$. Hence, we cannot have any privacy guarantees under this setting. Similar situations occur when we apply a random response mechanism on bits with value $0$. We also cannot prove the differential privacy property of the Bloom filter. Therefore, we have to apply the random response mechanism on bits either with value $0$ or $1$.

## 7 CONCLUSION AND FUTURE WORK

In this work, we introduced *DPBloomfilter*, a novel approach that leverages the random response mechanism to ensure the privacy of Bloom filters. To the best of our knowledge, this is the first work that applies random response to achieve differential privacy (DP) in the membership query tasks associated with Bloom filters.

From a privacy standpoint, we have rigorously demonstrated that our method achieves $(\epsilon, \delta)$-DP while retaining the same computational complexity as the standard Bloom filter. Furthermore, our theoretical analyses, complemented by extensive experimental evaluations, confirm that the DP-Bloomfilter not only upholds strong privacy guarantees but also maintains high utility.

Our results open several promising avenues for future research. In particular, one interesting direction is to explore more refined trade-offs between privacy and utility, potentially by further optimizing the random response mechanism to minimize any impact on accuracy. In summary, the DPBloomfilter integrates differential privacy into the Bloom filter data structure, and we anticipate our work can advance the state-of-the-art in privacy-preserving data structures.

ETHICS STATEMENT

This paper does not involve human subjects, personally identifiable data, or sensitive applications. We do not foresee direct ethical risks. We follow the ICLR Code of Ethics and affirm that all aspects of this research comply with the principles of fairness, transparency, and integrity.

REPRODUCIBILITY STATEMENT

We ensure reproducibility of our theoretical results by including all formal assumptions, definitions, and complete proofs in the appendix. The main text states each theorem clearly and refers to the detailed proofs. No external data or software is required.

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

# Appendix

**Roadmap.** The Appendix organizes as follows: In Section A, we introduce the notations used in the paper and differential privacy tools. In Section B, we elaborate the derivations for the closed-form distribution of the random variable $W$, where $N$ is the $1 - \delta$ quantile of $W$. Section C contains the proof of privacy guarantees for DPBloomfilter. Section D presents a detailed analysis of utility guarantees for DPBloomfilter. Section E restates the analysis results of running time for DPBloomfilter.

## A    BASIC TOOLS

In this section, we display the notations and basic tools for a better understanding of the readers. In Section A.1, we introduce the notations used in this paper. In Section A.2, we provide an essential basic composition Lemma for Differential Privacy.

### A.1    NOTATIONS

In this section, we describe the notations we use in this paper.

For any positive integer $n$, let $[n]$ denote the set $\{1, 2, \cdots, n\}$. We use $\mathbb{E}[]$ to denote the expectation operator and $\Pr[]$ to denote probability. We use $n!$ to denote the factorial of integer $n$. We use $A_m^n := \frac{m!}{(m-n)!}$ to denote the number of permutation ways to choose $n$ elements from $m$ elements considering the order of selection. We use $\binom{m}{n} := \frac{m!}{n!(m-n!)}$ to denote the number of combination ways to choose $n$ elements from $m$ elements without considering the order of selection. We use $F_X(x)$ to denote the Cumulative Distribution Function (CDF) of a random variable $X$ and use $F_X^{-1}(1 - \delta)$ to denote the $1 - \delta$ quantile of $F_X(x)$.

### A.2    BASIC COMPOSITION OF DIFFERENTIAL PRIVACY

If multiple differential privacy algorithms are involved, a composition rule becomes necessary. This section presents the simplest form of composition, as stated as follows:

**Lemma A.1** (Basic composition, (Ghazi et al., 2023)). *Let $M_1$ be an $(\epsilon_1, \delta_1)$-DP algorithm and $M_2$ be an $(\epsilon_2, \delta_2)$-DP algorithm. Then $M(X) = (M_1(X), M_2(M_1(X), X))$ is an $(\epsilon_1 + \epsilon_2, \delta_1 + \delta_2)$-DP algorithm.*

The basic composition lemma quantifies the total privacy loss across all operations. This is essential for determining whether the overall privacy guarantee remains acceptable.

## B    PROOF FOR $1 - \delta$ QUANTILE

In this section, we provide the calculation of the probability distribution of random variable $W := \sum_{j=1}^m \mathbb{1}\{g[j] \neq g'[j]\}$, which plays an important part in the proof of the privacy guarantee for our algorithm (see Section C). In Section B.1, we present the definition of random variables $W, Y, Z$ used in this section. In Section B.2, we calculate the probability distribution of $Y$. In Section B.3, we calculate the probability distribution of $Z$ conditioned on $Y$. In Section B.4, we calculate the probability distribution of $W$.

### B.1    DEFINITION

In this section, we present the definitions of random variables which will be used in the section.

**Definition B.1** (Definition of $W$). *Let $W := \sum_{j=1}^m \mathbb{1}\{g[j] \neq g'[j]\}$, where $g \in \{0, 1\}^m$ denotes the ground truth values generated by dataset $A$, and $g' \in \{0, 1\}^m$ denotes the ground truth values generated by neighboring dataset $A'$.*

**Definition B.2** (Definition of $Y$). *Consider a $x \in [n]$.*

*Let $y_1, y_2, \cdots, y_k$ denotes the $k$ hash values generated by the standard Bloom filter (Definition 3.1).*

*We define $Y$ as the set of distinct values among $y_1, y_2, \cdots, y_k$, where $|Y| \in 1, 2, \cdots, k$.*

**Definition B.3** (Definition of $Z$). *Consider two data $x, x' \in [n]$.*

*Let $y_1, y_2, \cdots, y_k$ denotes the $k$ hash values generated by $x$, and $y'_1, y'_2, \cdots, y'_k$ denotes the $k$ hash values generated by $x'$.*

*Follow the Definition B.2, let $Y_x$ denotes the set of distinct values in $y_1, y_2, \cdots, y_k$, and $Y_{x'}$ denotes the set of distinct values in $y'_1, y'_2, \cdots, y'_k$.*

*Suppose $|Y_x| = a, |Y_{x'}| = b$, where $a, b \in \{1, 2, \cdots, k\}$*

*We define $Z$ is the set of distinct values in $Y_x \cup Y_{x'}$, where $|Z| \in \{1, 2, \cdots, 2k\}$*

## B.2 Distribution of $Y$

Then we proceed to calculate the probability distribution of $Y$ in this section.

**Lemma B.4** (Distribution of $Y$). *If the following conditions hold*

- *Let $y_1, y_2, \cdots, y_k$ be defined in Definition B.2.*

- *Let $Y$ be defined as Definition B.2.*

*Then, we can show, for $y = 1, 2, \cdots, k$,*

$$\Pr[|Y| = y]$$
$$= \begin{cases} 1/m^{k-1}, & y = 1 \\ \binom{m}{y} \cdot y^k/m^k - \sum_{i=1}^{k-1} \binom{m-i}{y-i} \Pr[Y = i], & y = 2, \cdots, k \end{cases}$$

*Proof.* **Step 1.** We consider $Y = 1$ case.

Without any constraints, there are total $m^k$ situations. This is because each hash value can be freely chosen from $m$ positions, and there are $k$ hash values. Therefore, there are total $m^k$ situations.

Then, with constraint $Y = 1$, $k$ hash values must be assigned to the same position. The position can be chosen from a total of $m$ positions. Therefore, in this case, there are $m$ situations.

Combining the above two analysis, we have

$$\Pr[Y = 1] = \frac{m}{m^k}$$
$$= \frac{1}{m^{k-1}}.$$

**Step 2.** We consider $Y = 2, \cdots, k$ cases.

Similarly, without any constraints, there are total $m^k$ situations.

Since we need $Y = y$, we must choose $y$ from different positions in the total $m$ positions. Therefore, we have $\binom{m}{y}$ term.

Note that in each position, we need at least one hash value. We first compute the number of freely assigning $k$ hash values to the $y$ positions. Then we remove the failure cases.

As there are $y$ positions and $k$ hash values, we have the $y^k$ term for freely assigning $k$ hash values to $y$ positions.

For the failure case, we have $\sum_{i=1}^{k-1} \Pr[Y = i] \cdot \binom{m-i}{y-i}$. The $\binom{m-i}{y-i}$ term is due to repeated counting for each $i \in [k-1]$, where we first fix $i$ positions and then randomly pick the other $y - i$ different positions in the total $m - i$ positions.

Thus, in all, we have the following formula,

$$\Pr[Y = y] = \frac{\binom{m}{y} \cdot y^k}{m^k} - \sum_{i=1}^{k-1} \Pr[Y = i] \cdot \binom{m-i}{y-i}.$$

$\square$

## B.3 DISTRIBUTION OF $Z$ CONDITIONED ON $Y$

In this section, we calculate the probability distribution of $Z$ condition on $Y$.

**Lemma B.5** (Probability of $Z$ conditioned on $Y_x$ and $Y_{x'}$). *If the following conditions hold*

- *Let $Y_x, Y_{x'}, Z$ be defined as Definition B.3.*

- *Let $A_n^m$ denotes $n!/(n-m)!$.*

- *Let $t := z - \max(a, b)$.*

*Then, we can show, for $z = \max(a, b), \cdots, (a + b)$,*

$$\Pr[|Z| = z||Y_x| = a, |Y_{x'}| = b] = \frac{A_m^a \cdot \binom{b}{t} \cdot A_{m-a}^t \cdot A_a^{b-t}}{A_m^a \cdot A_m^b}.$$

*Proof.* Since the minimum value of $Z$ is $\max(a, b)$, without loss of generality, we assume $a \geq b$. Then we have $a \leq z \leq (a + b)$.

Recall we have $t = z - \max(a, b) = z - a, t \in \{0, 1, \cdots, b\}$. Then we have

$$\Pr[|Z| = a + t||Y_x| = a, |Y_{x'}| = b]$$
$$= \frac{A_m^a \cdot \binom{b}{t} \cdot A_{m-a}^t \cdot A_a^{b-t}}{A_m^a \cdot A_m^b}.$$

We explain why we have the above equation in the following steps.

**Step 1.** We consider the denominator.

Without any constraints, since $|Y_x| = a$, we need to choose $a$ from different positions in the total $m$ positions. Therefore, we have the $A_m^a$ term in the denominator. Similarly, since $|Y_{x'}| = b$, we have the $A_m^b$ term in the denominator.

**Step 2.** We consider the numerator.

Firstly, since $|Y_x| = a$, we need to choose $a$ different positions in total $m$ positions. Therefore, we have the $A_m^a$ term in the numerator.

Since $Z$ is defined as Definition B.3, we can have the following

$$|Y_x \cap Y_{x'}| = a + b - z$$
$$|Y_{x'}| - |Y_x \cap Y_{x'}| = z - a$$
$$= t$$

Then, we need to choose $t$ values from $Y_{x'}$ to construct $|Y_{x'}| - |Y_x \cap Y_{x'}|$ part. Therefore, we have the $\binom{b}{t}$ term in the numerator.

We also need to choose $t$ different positions in the rest $m - a$ positions for $|Y_{x'}| - |Y_x \cap Y_{x'}|$ part. Hence, we have the $A_{m-a}^t$ term in the numerator.

Lastly, let's consider the $b - t$ part. For this part, we need to choose $b - t$ different positions from $a$ positions. Therefore, we have the $A_a^{b-t}$ term in the numerator.

Combining all analyses together, finally, we have

$$\Pr[|Z| = z||Y_x| = a, |Y_{x'}| = b] = \frac{A_m^a \cdot \binom{b}{t} \cdot A_{m-a}^t \cdot A_a^{b-t}}{A_m^a \cdot A_m^b}.$$

$\square$

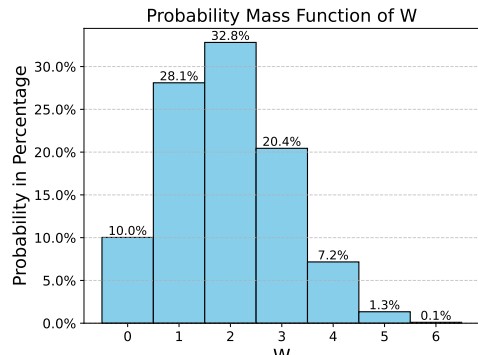

Figure 1: Let $W := |S|$ denote the number of bits in the Bloom filter changed by substituting an element in the inserted set $A$ (Definition 3.2). We achieve $\epsilon_0$-DP for each single bit and $(\epsilon, \delta)$-DP for the entire Bloom filter via the random response (Definition 3.4), where $\epsilon_0 = \epsilon/N$. The $N$ is $1 - \delta$ quantile of the random variable $W$. It can be inferred from this visualization that the values of random variable $W$ have good concentration properties, mostly concentrated around its mean.

### B.4 DISTRIBUTION OF $W$

Finally, we present the calculation of the probability distribution of $W$ in this section.

**Lemma B.6** (Distribution of $W$). *If the following conditions hold*

- *Let $Y_x, Y_{x'}, Z$ be defined as Definition B.3.*

- *Let $W$ be defined as Definition B.1.*

- *Let $A_n^m$ denotes $\frac{n!}{(n-m)!}$.*

- *Let $p_0 := (1 - \frac{1}{m})^{(|A|-1)k}$ denotes the proportion of bits which are still $0$ in the bit-array.*

- *Let $n_1 := |Y_x \cap Y_{x'}| = a + b - z$ denotes the number of overlap elements in $Y_x$ and $Y_{x'}$.*

- *Let $n_2 := |Y_x \cup Y_{x'}| - |Y_x \cap Y_{x'}| = z - (a + b - z) = 2z - a - b$ denotes the number of exclusive or elements in $Y_x$ and $Y_{x'}$.*

*Then, we can show, for $w = 0, \cdots 2k$,*

$$\Pr[W = w]$$
$$= \sum_{a=1}^{k} \sum_{b=1}^{k} \sum_{z=1}^{a+b} \Pr[W = w || Z| = z, |Y_x| = a, |Y_{x'}| = b]$$
$$\cdot \Pr[|Z| = z || Y_x| = a, |Y_{x'}| = b]$$
$$\cdot \Pr[|Y_x| = a] \cdot \Pr[|Y_{x'}| = b].$$

*where*

$$\Pr[W = w || Z| = z, |Y_x| = a, |Y_{x'}| = b]$$
$$= \begin{cases} 0, & n_2 < w \\ \binom{n_2}{w} \cdot p_0^w \cdot (1 - p_0)^{n_2 - w}, & n_2 \geq w \end{cases}$$

*Proof.* By basic probability rules, we have the following equation

$$\Pr[W = w]$$
$$= \sum_{a=1}^{k} \sum_{b=1}^{k} \sum_{z=1}^{a+b} \Pr[W = w || Z| = z, |Y_x| = a, |Y_{x'}| = b]$$

$$\cdot \Pr[|Z| = z||Y_x| = a, |Y_{x'}| = b]$$
$$\cdot \Pr[|Y_x| = a, |Y_{x'}| = b]$$
$$= \sum_{a=1}^{k}\sum_{b=1}^{k}\sum_{z=1}^{a+b}\Pr[W = w||Z| = z, |Y_x| = a, |Y_{x'}| = b]$$
$$\cdot \Pr[|Z| = z||Y_x| = a, |Y_{x'}| = b]$$
$$\cdot \Pr[|Y_x| = a] \cdot \Pr[|Y_{x'}| = b].$$

where the first step follows from basic probability rules, the second step follows from $Y_x$, and $Y_{x'}$ are independent.

We can get the probability of $\Pr[|Y_x| = a]$ and $\Pr[|Y_{x'}| = b]$ from Lemma B.4.

We can get the probability of $\Pr[|Z| = z||Y_x| = a, |Y_{x'}| = b]$ from Lemma B.5.

Now, let's consider the $\Pr[W = w||Z| = z, |Y_x| = a, |Y_{x'}| = b]$ term.

Note that only elements in the exclusive-or set may contribute to the final $W$. Therefore, we have $w \le n_2$. Namely, when $n_2 < w$, we have $\Pr[W = w||Z| = z, |Y_x| = a, |Y_{x'}| = b] = 0$.

Now, let's calculate $\Pr[W = w||Z| = z, |Y_x| = a, |Y_{x'}| = b]$ under $n_2 \ge w$ condition.

Recall $x$ denotes the element deleted from $A$, and $x'$ denotes the element added to $A$ for constructing the neighbor dataset $A'$.

Let $A_{fix} := A - x$ denote the fixed set of elements during the modifications. We have $|A_{fix}| = |A| - 1$.

Consider the following steps:

- We construct a new Bloom filter.

- We insert all elements in $A_{fix}$ in the Bloom filter.

- We define $Z_{zero}$ as the set of positions of bits which are still 0 after the insertion of $A_{fix}$.

We define $Z_{xor}$ as the exclusive-or set of $Y_x$ and $Y_{x'}$. We have

$$Z_{xor} = (Y_x \cup Y_{x'}) - (Y_x \cap Y_{x'}),$$
$$|Z_{xor}| = |Y_x \cup Y_{x'}| - |Y_x \cap Y_{x'}|$$
$$= z - (a + b - z)$$
$$= 2z - a - b$$
$$= n_2.$$

Note that only positions in $Z_{xor} \cap Z_{zero}$ will contribute to $W$. Namely, we need $|Z_{xor} \cap Z_{zero}| = w$.

We achieve the above condition by selecting $w$ elements in $Z_{xor}$ and let them satisfy the condition of $Z_{zero}$.

Therefore, we have

$$\Pr[|Z_{xor} \cap Z_{zero}| = w]$$
$$= \binom{n_2}{w} \cdot (1 - \frac{1}{m})^{(|A|-1)kw} \cdot (1 - (1 - \frac{1}{m})^{(|A|-1)k})^{n_2-w}.$$

Combining the above analysis, we have

$$\Pr[W = w||Z| = z, |Y_x| = a, |Y_{x'}| = b]$$
$$= \begin{cases} 0, & n_2 < w \\ \binom{n_2}{w} \cdot p_0^w \cdot (1 - p_0)^{n_2-w}, & n_2 \ge w \end{cases}.$$

$\square$

## C PRIVACY GUARANTEES FOR ONE COORDINATE

In this section, we provide proof of the privacy guarantees of the DPBloomfilter.

In Section C.1, we demonstrate the privacy guarantees for single bit of array in Bloom filter.

Then in Section C.2, we provide the proof of privacy guarantees for our entire algorithm.

### C.1 SINGLE BIT IS PRIVATE

We first consider the privacy guarantees of single bit of array in Bloom filter.

**Lemma C.1** (Single bit is private ). *If the following conditions hold:*

- *Let $\epsilon_0 \geq 0$.*

- *Let $\widetilde{g}[j] \in \{0, 1\}$ be the $i$-th element of array output by DPBloomfilter*

*Then, we can show that, for all $j \in [m]$, $\widetilde{g}[j]$ is $\epsilon_0$-DP.*

*Proof.* $\forall j \in [m]$, $g[j]$ is the ground truth value generated by dataset $A \subset [n]$. (An alternative view of $g$ is $g : [m] \to \{0, 1\}$.) Suppose $g[j] = u$, $u \in \{0, 1\}$. For any neighboring dataset $A' \subset [n]$, we denote the ground truth value generated by it as $g'[j]$. Similarly, we can define the $\widetilde{g}'[j]$.

We consider the following two cases to prove $\widetilde{g}[j]$ is $\epsilon_0$-DP, for all $j \in [m]$.

**Case 1.** Suppose $g'[j] = u$. We know

$$\Pr[\widetilde{g}[j] = u] = \frac{e^{\epsilon_0}}{e^{\epsilon_0} + 1},$$

$$\Pr[\widetilde{g}'[j] = u] = \frac{e^{\epsilon_0}}{e^{\epsilon_0} + 1}.$$

Combining the above two equations, then we obtain

$$\frac{\Pr[\widetilde{g}[j] = u]}{\Pr[\widetilde{g}'[j] = u]} = 1.$$

Similarly, we know

$$\Pr[\widetilde{g}[j] = 1 - u] = \frac{1}{e^{\epsilon_0} + 1},$$

$$\Pr[\widetilde{g}'[j] = 1 - u] = \frac{1}{e^{\epsilon_0} + 1}.$$

Combining the above two equations, then we obtain

$$\frac{\Pr[\widetilde{g}[j] = 1 - u]}{\Pr[\widetilde{g}'[j] = 1 - u]} = 1.$$

Thus, we know for all $v \in \{0, 1\}$,

$$\frac{\Pr[\widetilde{g}[j] = v]}{\Pr[\widetilde{g}'[j] = v]} = 1.$$

**Case 2.** Suppose $g'[j] \neq u$.

We know

$$\Pr[\widetilde{g}[j] = u] = \frac{e^{\epsilon_0}}{e^{\epsilon_0} + 1},$$

$$\Pr[\widetilde{g}'[j] = u] = \frac{1}{e^{\epsilon_0} + 1}.$$

Combining the above two equations, then we obtain

$$\frac{\Pr[\widetilde{g}[j] = u]}{\Pr[\widetilde{g}'[j] = u]} = e^{\epsilon_0}.$$

Similarly, we know

$$\Pr[\widetilde{g}[j] = 1 - u] = \frac{1}{e^{\epsilon_0} + 1},$$

$$\Pr[\widetilde{g}'[j] = 1 - u] = \frac{e^{\epsilon_0}}{e^{\epsilon_0} + 1}.$$

Combining the above two equations, then we obtain

$$\frac{\Pr[\widetilde{g}[j] = 1 - u]}{\Pr[\widetilde{g}'[j] = 1 - u]} = e^{-\epsilon_0}.$$

For $v \in \{0, 1\}$, we have

$$e^{-\epsilon_0} \leq \frac{\Pr[\widetilde{g}[j] = v]}{\Pr[\widetilde{g}'[j] = v]} \leq e^{\epsilon_0}.$$

Therefore, $\forall j \in [m]$, $\widetilde{g}[j]$ is $\epsilon_0$-DP. $\qquad\square$

### C.2 Privacy guarantees for DPBloomfilter

Then, we can prove that our entire algorithm is differentially private.

**Theorem C.2** (Privacy for Query, formal version of Lemma 4.1). *If the following conditions hold*

- *Let $N = F_W^{-1}(1 - \delta)$ denote the $1 - \delta$ quantile of the random variable $W$ (see Definition B.1).*

- *Let $\epsilon_0 = \epsilon/N$.*

*Then, we can show, the output of* QUERY *procedure of Algorithm 1 achieves $(\epsilon, \delta)$-DP.*

*Proof.* Let $A$ and $A'$ are neighboring datasets. Let $g \in \{0, 1\}^m$ is the ground truth value generated by dataset $A$, and $g' \in \{0, 1\}^m$ is the ground truth value generated by dataset $A'$.

We define

$$S := \{j \in [m] \ : \ g[j] \neq g'[j]\}.$$

We further define

$$\overline{S} := [m]\backslash S.$$

We consider two cases, **Case 1** is $j \in \overline{S}$ and **Case 2** is $j \in S$.

**Case 1**. $j \in \overline{S}$.

We can show that

$$\frac{\Pr[\widetilde{g}[j] = v]}{\Pr[\widetilde{g}'[j] = v]} = 1.$$

holds for $\forall v \in \{0, 1\}$.

**Case 2.** $j \in S$.

We can show that

$$e^{-\epsilon_0} \leq \frac{\Pr[\widetilde{g}[j] = v]}{\Pr[\widetilde{g}'[j] = v]} \leq e^{\epsilon_0}. \tag{2}$$

holds for $\forall v \in \{0, 1\}$.

Thus, for any $Z \in \{0, 1\}^m$, the absolute privacy loss can be bounded by

$$
\begin{aligned}
|\ln \frac{\Pr[\widetilde{g} = Z]}{\Pr[\widetilde{g'} = Z]}| &= |\ln \prod_{j \in S} \frac{\Pr[\widetilde{g}[j] = v]}{\Pr[\widetilde{g'}[j] = v]}| \\
&\leq |S|\epsilon_0 \\
&= |S|\frac{\epsilon}{N}.
\end{aligned}
\tag{3}
$$

where the first step follows from each entry of $g$ is independent, the second step follows from Eq. (2), and the last step follows from choice of $\epsilon_0$.

By the definition of $N$, we know that with probability at least $1 - \delta$, $|S| \leq F^{-1}(1 - \delta) = N$. Hence, Eq. (3) is upper bounded by $\epsilon$ with probability $1 - \delta$.

This proves the $(\epsilon, \delta)$-DP. $\qquad\square$

## D  UTILITY ANALYSIS

In this section, we establish the utility guarantees for our algorithm. Initially, we calculate the accuracy for the query of the standard Bloom filter in Section D.1. We then assess the utility loss caused by introducing the random response technique by comparing the output of the DPBloomfilter with the output of the standard Bloom filter in Section D.2. Ultimately, we present the assessment of our algorithm's utility in Section D.3.

We begin by defining the notation we will use in this section.

**Definition D.1.** *Let $z \in \{0, 1\}$ denote the true answer for whether $x \in A$. Let $\widehat{z} \in \{0, 1\}$ denote the answer outputs by* BLOOM FILTER*. Let $\widetilde{z} \in \{0, 1\}$ denote the answer output by* DPBLOOMFILTER *(Algorithm 1).*

### D.1  ACCURACY FOR QUERY OF STANDARD BLOOM FILTER

We first present the accuracy of the query of the standard bloom filter, as follows.

**Lemma D.2** (Accuracy for query of Standard Bloom Filter )*. If the following conditions hold*

- *Assume that a hash function selects each array position with equal probability.*

- *Let $\widehat{z}$ be defined as Definition D.1.*

- *Let $z$ be defined as Definition D.1.*

- *Let $\alpha := \Pr[z = 0]$*

*Then, we can show*

$$\Pr[\widehat{z} = z] \geq 1 - (1 - e^{-2|A|k/m})^k \cdot \alpha.$$

*Further if $m = \Omega(|A|k)$ and $k = \Theta(\log(1/\delta_{err}))$, we have*

$$\Pr[\widehat{z} = z] \geq 1 - \delta_{err} \cdot \alpha.$$

*Proof.* Recall that we have defined Bloom filter in Definition 3.1, it only has false positive error. Therefore, we only need to calculate the following

$$\Pr[\widehat{z} = 1 | z = 0]$$

Recall that $A \subset [n]$ denotes the set of elements inserted into the Bloom filter. And $h_i : [n] \to [m]$ for each $i \in [k]$ denotes $k$ hash functions used in the Bloom filter.

For a query $y \notin A$, we denotes event $E_1$ happens if the following happens:

$$h_i[y] = 1, \forall i \in [k]$$

Recall that we have defined Bloom filter in Definition 3.1, we have

$$\Pr[\widehat{z} = 1 | z = 0] = \Pr[E_1]. \tag{4}$$

Now, we start calculating $\Pr[E_1]$.

Recall that we assume a hash function selects each array position with equal probability in the lemma statement.

During one inserting operation, the probability of a certain bit is not set to 1 is

$$(1 - \frac{1}{m})^k$$

If we have inserted $|A|$ elements, the probability that a certain bit is still 0 is

$$(1 - \frac{1}{m})^{|A|k} = ((1 - \frac{1}{m})^m)^{|A|k/m} \geq e^{-2|A|k/m}$$

where the last step follows from $(1 - 1/m)^m \geq e^{-2}$ for all $m \geq 2$.

Thus the probability that a certain bit is 1 is

$$1 - (1 - \frac{1}{m})^{|A|k} \leq 1 - e^{-2|A|k/m}.$$

Combining the above fact, we have

$$\Pr[E_1] = (1 - (1 - \frac{1}{m})^{|A|k})^k$$
$$\leq (1 - e^{-2|A|k/m})^k. \tag{5}$$

where the first step follows from the definition of event $E_1$, the second step follows from $(1 - 1/m)^m \geq e^{-2}$ for all $m \geq 2$.

Therefore, the accuracy of Bloom filter is

$$\Pr[\widehat{z} = z] = 1 - \Pr[\widehat{z} = 1 | z = 0] \Pr[z = 0]$$
$$= 1 - \Pr[E_1]\alpha$$
$$\geq 1 - (1 - e^{-2|A|k/m})^k \alpha.$$

where the first step follows from Bloom filter only has false positive error, the second step follows from the definition of event $E_1$ and the definition of $\alpha$, the third step follows from Eq. (5).

$\square$

### D.2 ACCURACY (COMPARE DPBLOOMFILTER WITH STANDARD BLOOMFILTER) FOR QUERY

We then assess the accuracy loss caused by the introduction of the random response technique by comparing the outputs of the DPBloomfilter with those of the standard Bloom filter.

**Lemma D.3** (Accuracy (compare DPBloomFilter with Standard BloomFilter) for Query ). *If the following conditions hold*

- *Let $\widehat{z}$ be defined as Definition D.1.*

- *Let $\widetilde{z}$ be defined as Definition D.1.*

- *Let $\alpha := \Pr[z = 0] \in [0, 1]$*

- *Let $t := \frac{e^{\epsilon_0}}{e^{\epsilon_0} + 1}$.*

- *Let $\delta_{\mathrm{err}}$ be defined as in Lemma D.2.*

*Then, we can show*

$$\Pr[\widetilde{z} = \widehat{z}] \geq t \cdot (\alpha - \delta_{\mathrm{err}}).$$

*Proof.* We denote the query as $q$.

We define

$$Q := \{j \in [m] \ : \ h_i(q) = j, \ i \in [k]\} \tag{6}$$

We denote $Q[i]$ as the $i$-th element in $Q$.

Using basic probability rules, we have

$$\begin{aligned}
&\Pr[\widetilde{z} = \widehat{z}] \\
&= \Pr[\widetilde{z} = 1|\widehat{z} = 1]\Pr[\widehat{z} = 1] \\
&+ \Pr[\widetilde{z} = 0|\widehat{z} = 0]\Pr[\widehat{z} = 0].
\end{aligned}$$

**Step 1**. Calculate $\Pr[\widetilde{z} = 1|\widehat{z} = 1]$

We denote event $E_2$ happens as the following happens:

$$\widetilde{g}[j] = g[j], \forall j \in Q.$$

Recall that we have defined Bloom filter in Definition 3.1, we have

$$\Pr[\widetilde{z} = 1|\widehat{z} = 1] = \Pr[E_2].$$

Now, we calculate the probability that $E_2$ happens.

$$\begin{aligned}
\Pr[E_2] &= \prod_{i=1}^{k} \Pr[\widetilde{g}[Q[i]] = g[Q[i]]] \\
&= (\frac{e^{\epsilon_0}}{e^{\epsilon_0} + 1})^k.
\end{aligned}$$

where the first step follows from each entry of $g$ is independent, the second steps follows from the definition of $\widetilde{g}$.

Therefore, we have

$$\Pr[\widetilde{z} = 1|\widehat{z} = 1] = (\frac{e^{\epsilon_0}}{e^{\epsilon_0} + 1})^k. \tag{7}$$

**Step 2**. Calculate $\Pr[\widetilde{z} = 0|\widehat{z} = 0]$

Recall we have defined $Q \subset [m]$ in Eq. (6). We further define

$$Z := \{j \in Q \ : \ g[j] = 0\}.$$

We denote $Z[i]$ as the $i$-th element in $Z$.

We further define

$$\overline{Q} := Q \backslash Z.$$

By basic probability rules, we have

$$\Pr[\widetilde{z} = 0|\widehat{z} = 0] = 1 - \Pr[\widetilde{z} = 1|\widehat{z} = 0].$$

Now, let's calculate $\Pr[\widetilde{z} = 1|\widehat{z} = 0]$

$[\widetilde{z} = 1|\widehat{z} = 0]$ happens only if the following conditions hold:

1. All elements in $Z$ flip from $0$ to $1$.

2. All elements in $\overline{Q}$ remain $1$.

Then, we have

$$
\begin{aligned}
\Pr[\widetilde{z} = 1 | \widehat{z} = 0] &= \prod_{i=1}^{|Z|} \Pr[\widetilde{g}[Z[i]] = 1] \prod_{i=1}^{|\overline{Q}|} \Pr[\widetilde{g}[\overline{Q}[i]] = 1] \\
&= (\frac{1}{e^{\epsilon_0} + 1})^{|Z|} (\frac{e^{\epsilon_0}}{e^{\epsilon_0} + 1})^{|\overline{Q}|} \\
&\leq (\frac{1}{e^{\epsilon_0} + 1})^{|Z|} \\
&\leq \frac{1}{e^{\epsilon_0} + 1}.
\end{aligned}
$$

where the first step follows from the above analysis, the second step follows from the definition of $\widetilde{g}$, the third step follows from $|\overline{Q}| \geq 0$ and $\frac{e^{\epsilon_0}}{e^{\epsilon_0}+1} < 1$, the fourth step follows from $|Z| \geq 1$ and $\frac{1}{e^{\epsilon_0}+1} < 1$.

Therefore, we have

$$
\begin{aligned}
\Pr[\widetilde{z} = 0 | \widehat{z} = 0] &= 1 - \Pr[\widetilde{z} = 1 | \widehat{z} = 0] \\
&\geq 1 - \frac{1}{e^{\epsilon_0} + 1} \\
&= \frac{e^{\epsilon_0}}{e^{\epsilon_0} + 1}. \tag{8}
\end{aligned}
$$

Let $\widehat{\alpha} := \Pr[\widehat{z} = 0]$, then we have $1 - \widehat{\alpha} = \Pr[\widehat{z} = 1]$. Let $\alpha := \Pr[z = 0]$. Note that $\widehat{\alpha} = \alpha(1 - \delta_{\mathrm{err}})$.

Let $t := \frac{e^{\epsilon_0}}{e^{\epsilon_0}+1}$.

The final accuracy is

$$
\begin{aligned}
&\Pr[\widetilde{z} = 0 | \widehat{z} = 0] \cdot \Pr[\widehat{z} = 0] + \Pr[\widetilde{z} = 1 | \widehat{z} = 1] \cdot \Pr[\widehat{z} = 1] \\
&= \Pr[\widetilde{z} = 0 | \widehat{z} = 0] \cdot \widehat{\alpha} + \Pr[\widetilde{z} = 1 | \widehat{z} = 1] \cdot (1 - \widehat{\alpha}) \\
&= \Pr[\widetilde{z} = 0 | \widehat{z} = 0] \cdot \alpha(1 - \delta_{err}) \\
&\quad + \Pr[\widetilde{z} = 1 | \widehat{z} = 1] \cdot (1 - \alpha + \alpha \cdot \delta_{err}) \\
&\geq \frac{e^{\epsilon_0}}{e^{\epsilon_0} + 1} \cdot \alpha(1 - \delta_{err}) + (\frac{e^{\epsilon_0}}{e^{\epsilon_0} + 1})^k \cdot (1 - \alpha + \alpha \cdot \delta_{err}) \\
&= t \cdot (\alpha - \alpha \cdot \delta_{err}) + t^k \cdot (1 - \alpha + \alpha \cdot \delta_{err}) \\
&\geq t \cdot \alpha \cdot (1 - \delta_{err}).
\end{aligned}
$$

where the first step follows from the definition of $\widehat{\alpha}$, the second step follows from $\widehat{\alpha} = \alpha(1 - \delta)$, the third step follows from Eq. (7) Eq. (8), the fourth step follows from basic algebra rules, the fifth step follows from $(1 - \alpha + \alpha \cdot \delta_{\mathrm{err}}) \geq 0$.

Therefore, the final accuracy is $t \cdot (\alpha - \delta_{err})$. $\qquad\square$

### D.3 ACCURACY (COMPARE DPBLOOMFILTER WITH TRUE-ANSWER) FOR QUERY

Now we can examine the utility guarantees of DPBloomfilter by calculating the error between the ground truth for query and the output of DPBloomfilter.

**Theorem D.4** (Accuracy (compare DPBloomfilter with true-answer) for Query, formal version of Lemma 4.2). *If the following conditions hold*

- *Let $\widehat{z}$ be defined as Definition D.1.*

- *Let $z$ be defined as Definition D.1.*

- *Let $\alpha := \Pr[z = 0] \in [0, 1]$*

- *Let $t := e^{\epsilon_0}/(e^{\epsilon_0} + 1)$.*

- *Let $\delta_{\mathrm{err}}$ be defined as in Lemma D.2.*

*Then, we can show*

$$\Pr[\widetilde{z} = z] \geq \alpha(1 - t - t^k)\delta_{\mathrm{err}} + \alpha t.$$

*Proof.* We have

$$
\begin{aligned}
\Pr[\widetilde{z} &= z] \\
&= \Pr[\widetilde{z} = 0|\widehat{z} = 0]\Pr[\widehat{z} = 0|z = 0]\Pr[z = 0] \\
&+ \Pr[\widetilde{z} = 0|\widehat{z} = 1]\Pr[\widehat{z} = 1|z = 0]\Pr[z = 0] \\
&+ \Pr[\widetilde{z} = 1|\widehat{z} = 1]\Pr[\widehat{z} = 1|z = 1]\Pr[z = 1] \\
&+ \Pr[\widetilde{z} = 1|\widehat{z} = 0]\Pr[\widehat{z} = 0|z = 1]\Pr[z = 1] \\
&\geq t \cdot (1 - \Pr[E_1]) \cdot \alpha + (1 - t^k) \cdot \Pr[E_1] \cdot \alpha + t^k \cdot 1 \cdot (1 - \alpha) \\
&= \alpha(1 - t - t^k)\delta_{\mathrm{err}} + \alpha t + t^k(1 - \alpha) \\
&\geq \alpha(1 - t - t^k)\delta_{\mathrm{err}} + \alpha t.
\end{aligned}
$$

where the first step from basic probability rules, the secod step follows from Equation 4, Equation 8 and definition of $\alpha$ and $t$, the third step follows from basic algebra, the fourth step follows from the fact that $t, \alpha \in [0, 1]$.

$\square$

To make it easier to understand, we also provide the utility analysis of the Bloom filter under the case of random guess.

**Lemma D.5** (Accuracy for Query under Random Guess). *If the following conditions hold*

- *Let $\widehat{z}$ be defined as Definition D.1.*

- *$\epsilon_0 = 0$. Namely, each bit in the bit-array of the DP Bloom has $\frac{1}{2}$ probability to be set to $0$, and $\frac{1}{2}$ probability to be set to $1$.*

*Then, we can show*

$$\Pr[\widetilde{z} = 0] = 1 - \frac{1}{2^k},$$

$$\Pr[\widetilde{z} = 1] = \frac{1}{2^k}.$$

*Proof.* By the definition of Bloom filter 3.1, the answer $\widetilde{z} = 1$ requires $k$ corresponding positions in the bit-array of the query are all set to $1$.

Note that each bit has $\frac{1}{2}$ probability to be set to $1$. Therefore, we have

$$\Pr[\widetilde{z} = 1] = \frac{1}{2^k}.$$

Then, we have $\Pr[\widetilde{z} = 0] = 1 - \Pr[\widetilde{z} = 1] = 1 - \frac{1}{2^k}$.

$\square$

## E  RUNNING TIME

In this section, we provide the proof of running time for Algorithm 1. The running time for our algorithm consists of two parts: time for initialization in Section E.1 and time for query in Section E.2.

### E.1 Running time for initialization

Now we calculate the time of initialization for our algorithm.

**Lemma E.1** (Running time for initialization). *Let $\mathcal{T}_h$ denote the time of evaluation of function $h$ at any point.*

*It takes $O(|A| \cdot k \cdot \mathcal{T}_h + m)$ time to run the initialization function.*

*Proof.* **Step 1** Let's consider the initialization of the standard Bloom filter.

A single element $x$ needs $O(k \cdot \mathcal{T}_h)$ time to compute over $k$ hash functions.

There are $|A|$ elements which need to be inserted.

Combining the above two facts, it needs $O(|A| \cdot k \cdot \mathcal{T}_h)$ time to initialise the standard Bloom filter.

**Step 2** Let's consider the "Flip each bit" part.

Since there are $m$ bits in the Bloom filter, it takes $O(m)$ time to flip each bit.

Therefore, the initialization function needs $O(|A| \cdot k \cdot \mathcal{T}_h + m)$ time to run.

$\square$

### E.2 Running time for query

Then, we proceed to calculate the query time for our algorithm.

**Lemma E.2** (Running time for query). *Let $\mathcal{T}_h$ denote the time of evaluation of function $h$ at any point. It takes $O(k \cdot \mathcal{T}_h)$ time to run each query $y$ in the query function.*

*Proof.* For each query $y$, the algorithm needs $O(k \cdot \mathcal{T}_h)$ time to compute the hash values of $y$ over $k$ hash functions.

Therefore, it takes $O(k \cdot \mathcal{T}_h)$ time to run the query function for each query. $\square$

By combing the result of Lemma E.1 and Lemma E.2, we can obtain the running of our entire algorithm is $O(|A| \cdot k \cdot \mathcal{T}_h + m)$.

## LLM Usage Disclosure

LLMs were used only to polish language, such as grammar and wording. These models did not contribute to idea creation or writing, and the authors take full responsibility for this paper's content.

