# OpenReview forum: "DPBloomfilter: Securing Bloom Filters with Differential Privacy"
_ICLR.cc/2026/Conference — ICLR 2026 Conference Withdrawn Submission_

### Official Review · Reviewer_VWRm · 2025-10-21

**Soundness:** 2
**Presentation:** 2
**Contribution:** 1
**Rating:** 2
**Confidence:** 4

**Summary:**

The paper proposes DPBloomfilter, which integrates a differential privacy (DP) mechanism, specifically randomized response, into a Bloom filter. The approach flips each bit in the Bloom filter with some probability to achieve an ((\varepsilon, \delta))-DP guarantee. The authors show that the resulting structure retains the same computational complexity as a standard Bloom filter, and they provide theoretical proofs of privacy and basic bounds on utility.

**Strengths:**

The paper is reasonably well-written, and it provides a mathematical analysis of the randomized response composition.
It applies a theoretical framework that is consistent and technically sound.

**Weaknesses:**

The paper does the most basic thing one might expect for adding differential privacy to a Bloom filter, perturbing bits independently via randomized response. The analysis follows straightforwardly from standard DP composition and doesn’t introduce conceptual or algorithmic novelty beyond that. Consequently, while technically correct, the work is not particularly original or deep. Indeed, as the authors note, this idea was examined in the BLIP paper from 2012, and I see the difference between this paper and that paper as being minimal.

The authors also note that introducing this type of noise inevitably leads to false negatives. The main advantage of a Bloom filter, however, is precisely that it avoids false negatives. Once false negatives are allowed, there is little reason to start from a Bloom filter in the first place. Other approximate membership or sketching structures (e.g. quotient or cuckoo filters, locality-sensitive hashing structures) may allow both false positives and false negatives more naturally. A discussion of these alternatives, or a comparison against data structures designed to tolerate symmetric error, is missing. The authors should acknowledge that this changes the underlying semantics of membership queries and that the resulting structure is not a “DP Bloom filter” so much as a “DP approximate membership sketch.” It is less clear what are the right applications for a DP approximate membership sketch.

I don't feel the paper was clearly written regarding what the DP meant in the context of a Bloom filter.  My understanding would be “neighboring” datasets (Bloom filters for sets (S) and (S') differing by one element) would be the relevant starting point, and the privacy claim would be the results for such 2 sets would be what was considered for DP. That is probably what the authors are doing, but they should make it clearer? The meaning of the DP guarantee in the context of probabilistic membership queries seemed to me underspecified.

Finally, the paper does not engage with a significant body of prior work on secure or private variants of Bloom filters that are not based on differential privacy but pursue similar goals. Examples include:
https://arxiv.org/pdf/2501.15751
https://www.tdp.cat/issues/tdp.a015a09.pdf

**Questions:**

Please clarify the DP definition and semantics as they are applied to Bloom filters. Explicitly state what the neighboring datasets are and what privacy claim the mechanism guarantees for membership queries.

Since false negatives negate one of the main advantages of Bloom filters, what are the specific use cases where the DP variant remains useful?  Can you show that your proposal is useful in these cases directly?

Are there other noise approaches one could add besides the basic randomized response?

---

> ### Author Response · Authors · 2025-11-27
>
> Thank you for your thoughtful feedback. Your comments are very helpful and much appreciated. We will address these in the next version.

---

### Official Review · Reviewer_tVyW · 2025-10-24

**Soundness:** 1
**Presentation:** 2
**Contribution:** 1
**Rating:** 2
**Confidence:** 4

**Summary:**

The paper studies membership queries under differential privacy. In this problem, a database is a set of items from a known domain, and the goal is to output a data structure that can take a new item $x$ and answer whether or not it was in the original set. The paper proposes a solution combining Bloom filters and randomized response.

**Strengths:**

The problem is natural, and a bloom filter is an intuitive direction. Improving the privacy analysis by directly analyzing the filter's output distribution instead of using basic composition could also be a nice step.

**Weaknesses:**

1) The paper doesn't provide any evidence for the quality of its solution. Possible evidence would be experimental comparison to existing methods (see, e.g., Patel et al 24, https://openreview.net/pdf?id=GQNvvQquO0 for one method, as well as others that it discusses) or a comparison of formal utility guarantees (which the aforementioned paper also has). Briefly, what does this method provide that the others don't?

2) The paper lacks a clear (or even, technically, correct -- see Questions below) statement of how to translate a desired $(\varepsilon, \delta)$-DP guarantee into the parameter setting $\varepsilon$. This makes it hard for a reader to actually use the algorithm in practice, or even understand what the asymptotic relationship between the overall privacy guarantee and utility is.

**Questions:**

1) In Step 2 of the proof of Lemma B.4, the paper claims that $P[|Y| = y] = \binom{m}{y}(y/m)^k - \sum_{i=1}^{k-1} \binom{m-i}{y-i}P[Y=i]$. First, I think that's supposed to be $P[|Y|=i]$ at the end, since $Y$ is a set, not a number. Second, should the sum be to $y-1$ rather than $k-1$? If $k > y$, for example, this is circular. I don't think this actually changes most of the paper, but that is also kind of a problem -- it's hard to actually fish out the explicit values for parameters like $\varepsilon_0$ that are based on things like $F_W^{-1}(1-\delta)$ that are not explicitly provided anywhere. This calls statements like "[w]e have proved from a theoretical perspective that when the DP parameters $\varepsilon$ and $\delta$ are very small, DPBloomfilter can still maintain good utility" (Line 75) into question, as the paper doesn't actually say what the relationship between $(\varepsilon, \delta)$ and $\varepsilon_0$ is, but states its utility guarantees in terms of $\varepsilon_0$.

2) The abstract (Line 18), introduction (Line 76), and conclusion (Line 480) all mention experiments, but I don't see any experiments. The paper should at least compare to the Patel et al paper (https://openreview.net/pdf?id=GQNvvQquO0) and the basic composition bloom filter from Alaggan et al.

3) Similarly, the abstract and intro advertise that the paper is "the first work to provide DP for the Bloom filter for membership query problems", but then clarify that Alaggan et al did the same thing in 2012 (albeit for pure, not approximate DP). This should be rewritten.

4) Lemma 5.4 has RHS $t\alpha(1-\delta_{err})$, but Lemma D.3 variously says $t(\alpha - \delta_{err})$ and $t\alpha(1-\delta_{err})$.

5) Under Condition 5.2, only Condition 1 is really a condition. The rest appear to just be definitions and should be labelled accordingly. A similar comment applies to Lemma B.5 (and maybe other places I didn't catch).

6) What does "[i]t implements the heretical storage form of hash value atmosphere quotient and remainder" (Line 89) mean?

---

> ### Author Response · Authors · 2025-11-27
>
> Thank you for your thoughtful feedback. Your comments are very helpful and much appreciated. We will address these in the next version.

---

### Official Review · Reviewer_g6NE · 2025-10-31

**Soundness:** 2
**Presentation:** 2
**Contribution:** 2
**Rating:** 4
**Confidence:** 3

**Summary:**

This paper introduces the concept of differential privacy and combines it with the existing Bloom Filter method to design DPBloom Filter, a sketch that solves the membership query problems while satisfying $\epsilon$-differentially privacy. The author proves the differential privacy guarantees for each single bit of DPBloom filter as well as for the algorithm. The author also states that the algorithm does not increase too many errors under certain circumstances.

**Strengths:**

1. The paper introduces a new problem, privacy protection in sketches. This is a creative combination of existing ideas and has great connection with reality.
2. The paper presents a very detailed mathematical analysis of the theoretical properties of the algorithm.

**Weaknesses:**

1. The algorithm presented in the paper is too simple and has important defects. The original Bloom Filter produces only negative errors, but this algorithm also produced positive errors, which is a big problem.
2. There is no experimental results, which makes me doubt the claim that the algorithm "can still maintain good utility".

**Questions:**

1. One important feature of the Bloom filter is that it produces one-sided error. Can you try to adjust your algorithm so that it maintains this feature?
2. I would suggest conducting an experiment to compare the accuracy of your algorithm with the original Bloom Filter and other sketches that solves membership query problems? If you want to persuade people that your algorithm does not produce too much error you need experiment result to illustrate this.
3. The theoretical part seems a bit overly exhaustive to me. Can you remove some trivial parts and leave some space for experiments and diagrams?

---

> ### Author Response · Authors · 2025-11-27
>
> Thank you for your thoughtful feedback. Your comments are very helpful and much appreciated. We will address these in the next version.

---

### Official Review · Reviewer_Ma7S · 2025-11-05

**Soundness:** 1
**Presentation:** 3
**Contribution:** 1
**Rating:** 2
**Confidence:** 4

**Summary:**

This paper proposes a differentially-private variant of Bloom filter, which is a well-known data structure used for performing membership queries. More precisely, after being built each bit of the Bloom filter is randomized to ensure (epsilon, delta)-differential privacy.

**Strengths:**

-The paper is well-written and the authors have clearly reviewed the related work on Bloom filters and their wide range of use. The background notions on Bloom filters and differential privacy are also clearly summarized.

-The authors have provided a detailed theoretical analysis of the accuracy of the mechanism.

**Weaknesses:**

-The approach has not been validated experimentally or compared to other state-of-the-art approaches such as BLIP. Thus, the theoretical analysis has not been validated experimentally.

-There is clear tension between maintaining the accuracy of membership queries vs protecting the privacy of elements that have been inserted in the Bloom filter. In particular, if the accuracy of membership queries is high this means that an adversary can perform a reconstruction attack simply by enumerating potential items that are stored in the Bloom filter and he will be able to reconstruct a faithful list of items inserted due to the accuracy guarantees. This is a fundamental trade-off that should be discussed further in the privacy analysis. In particular, other constructions such as BLIP and RAPPOR are usually used for performing privacy-preserving analytics rather than membership queries.

-The novelty is rather limited compared to the previous work of Alaggan et al. 2012 on the BLIP construction, which was already focusing on achieving epsilon-differential privacy for Bloom filter.

Minor typos :
-« a random response mechanism » -> « a randomized response mechanism »
-« the random response mechanism » -> « the randomized response mechanism »

**Questions:**

Please see the main points raised in the weaknesses section.

---

> ### Author Response · Authors · 2025-11-27
>
> Thank you for your thoughtful feedback. Your comments are very helpful and much appreciated. We will address these in the next version.

---

### Note · Authors · 2025-11-27

**Comment:**

We would like to sincerely thank all the reviewers for providing constructive feedback on our paper. We will carefully address these comments in our next version. After careful consideration, we have decided to withdraw this paper.

**Withdrawal Confirmation:**

I have read and agree with the venue's withdrawal policy on behalf of myself and my co-authors.